# FACAM: A Fast and Accurate Clustering Analysis Method for Protein Complex Quantification in Single Molecule Localization Microscopy

**Cheng Wu [1], Weibing Kuang [2], Zhiwei Zhou [2], Yingjun Zhang [1,*] and Zhen-Li Huang [1]**

[1] Key Laboratory of Biomedical Engineering of Hainan Province, School of Biomedical Engineering, Hainan University, Haikou 570228, China

[2] Britton Chance Centerand MoE Key Laboratory for Biomedical Photonics, School of Engineering Sciences, Wuhan National Laboratory for Optoelectronics, Huazhong University of Science and Technology, Wuhan 430074, China

[*] Correspondence: yjzhang@hainanu.edu.cn

**Abstract:** Single molecule localization microscopy (SMLM) enables the analysis and quantification of protein complexes at the nanoscale. Using clustering analysis methods, quantitative information about protein complexes (for example, the size, density, number, and the distribution of nearest neighbors) can be extracted from coordinate-based SMLM data. However, since a final super-resolution image in SMLM is usually reconstructed from point clouds that contain millions of localizations, current popular clustering methods are not fast enough to enable daily quantification on such a big dataset. Here, we provide a fast and accurate clustering analysis method called FACAM, which is modified from the Alpha Shape method (a point dataset analysis method used in many fields). By taking advantage of parallel computation, FACAM is able to process millions of localizations in less than an hour, which is at least 10 times faster than the popular DBSCAN method. Furthermore, FACAM adaptively determines the segmentation threshold, and thus overcomes the problem of user-defined parameters. Using simulation and experimental datasets, we verified the advantages of FACAM over other reported clustering methods (including Ripley's H, DBSCAN, and ClusterViSu).

**Keywords:** single molecule localization microscopy; quantitative cluster analysis; big dataset; alpha shape

## 1. Introduction

In recent years, single molecule localization microscopy (SMLM) has opened up a new window for studying important questions in cell biology [1], such as gene transcription [2] or inter-organelle contacts [3]. Because SMLM achieves a spatial resolution of typically 20–30 nm in biological samples, we are now able to investigate important information about the molecular structure and interactions of protein complexes at the nanoscale [4]. Unlike traditional optical microscopy techniques that rely on grey scale images, after applying single molecular localization to thousands of raw images, SMLM obtains a coordinate-based localization table, and uses this table to reconstruct a final super-resolution image. Therefore, it is widely accepted that performing quantification on the localization table is better than that on the reconstructed super-resolution image [4]. However, it is important to point out that, to make the quantification meaningful and reliable, it is preferred to perform clustering on the localization coordinates [5], so that certain characteristics (for example, the structure of the nuclear pore complex) [6] of the localizations can be better observed.

Researchers have been developing different clustering methods for SMLM data, and some of them have been popularly used in the SMLM community. For example, the Ripley function, which is based on second-order statistics [7], has been used to analyze protein

organization by performing a statistical analysis on spatial points [8,9], so that a global overview of cluster characteristics can be obtained. However, this method is not able to handle heterogeneous protein clusters. More importantly, the Ripley function cannot provide accurate information about the number, location, or shape of protein clusters [5]. Another clustering method, called density-based spatial clustering of applications with noise (DBSCAN [10]), which has been applied to different applications [11–13], has good noise immunity, but requires setting two parameters (MinPts and $\varepsilon$) manually. Furthermore, unfortunately, the performance of DBSCAN varies significantly for different parameter settings [14]. Some methods [14,15] have been used to solve the problem of parameter selection for DBSCAN, however, none of them optimize the clustering speed of DBSCAN. The grid-based clustering method FOCAL [16], which achieves faster clustering than DBSCAN, still requires a user-defined parameter (minL). Recently, Voronoi-based clustering methods, including ClusterViSu [17] and SR-Tesseler [5], have been developed to solve the manual setting problem; however, they may face the segmentation issue when the data are non-isotopically distributed [18]. The graph-based method (StormGraph [19]) can process big data quickly, but its clustering results are highly dependent on the graph construction method and parameters (k).

It is worth pointing out that, a localization table contains not only the coordinates of localizations, but also other important molecule information (especially localization precision). Unfortunately, most of the current coordinate-based clustering analysis methods consider only the coordinates, and do not pay attention to the localization precision [18]. More importantly, the reported clustering methods may be too slow to be used daily when a localization table contains millions of localizations. Note that such a table size is common in daily SMLM experiments. Specifically, the Bayesian methods [20] would take about 19 h for a dataset with 30 small regions of interest (ROIs), which is unacceptable for daily use. For DBSCAN (MinPts = 5, $\varepsilon$ = 10 nm) and the Ripley function, we found out that they required about 7 h and 3 h, respectively, to process one million localizations. All programs were run on a desktop computer with an Intel i7-9700 CPU at 3.00GHz and 16GB of RAM. Furthermore, as a representative Voronoi-based clustering method, ClusterViSu can only analyze ROIs where the number of localizations does not exceed the maximum, which is dependent on the computer's RAM. This is because ClusterViSu uses the "pdist2 function" of MATLAB, which has a maximum allowable array size depending on the size of the computer memory. Therefore, after considering all issues mentioned above, we conclude that it is necessary to develop fast clustering analysis methods for processing a localization table that is large enough to match the size of daily SMLM data.

Here, after several modifications to the Alpha Shape method (a point dataset analysis method popularly used in different research fields) [21], we propose a fast clustering analysis method called FACAM (a fast and accurate clustering analysis method for protein complex quantification). Unlike most clustering analysis methods that rely only on the coordinates of localizations, FACAM employs both the coordinates and the localization precision in the clustering, and the latter is used to achieve an adaptive determination of segmentation thresholds. We verified that, after parallel computation acceleration, FACAM is able to process millions of localizations in half an hour. We compared the data processing speed of FACAM with other reported clustering methods (including ClusterViSu, DBSCAN, and Ripley's H), and proved that FACAM runs at least 10 times faster than DBSCAN. We demonstrated the capability of applying FACAM on different biological structures, including membrane proteins, erythrocyte cytoskeletons, and nuclear pore complexes.

## 2. Methods

### 2.1. Experimental Datasets

Firstly, we obtained the erythrocyte cytoskeleton dataset (the N termini of β-spectrin) from Xu lab [22]. This STORM dataset is a localization table that contains information about coordinates and localization precision. Secondly, we downloaded the nuclear pore complex (Nup96-GEP nanobody-X4-AF647) dataset from the supported information in

Thevathasan's paper [6]. This SMLM dataset contains 9 localization tables. Each localization table contains about $1 \times 10^5$ localizations. Thirdly, we collected the SMLM datasets of monomers (CD86-mEos4b) and dimers (CTLA4-mEos4b) from Baldering's paper [23]. This PALM dataset contains 10 monomers and 10 dimers. Finally, since the CD138 data were from our group, we performed SMLM experiments on membrane protein CD138 using the procedures described below. All the localization tables we used contain information about the spatial coordinates and localization precision of each localization.

### 2.1.1. Cell Culture and Sample Preparation

RPMI-8226 cells (purchased from Boster Biological Technology, Wuhan) were grown in RPMI-1640 medium supplemented with Fetal Bovine Serum (10%), penicillin (100 U/mL), and streptomycin (0.1 mg/mL) at 37 °C and 5% $CO_2$. Glass bottom dishes (MatTek P35G-1.5-14-C) were coated with 0.1 mg/mL poly-L-lysine for 3 h at room temperature and dried at 37 °C. Then, 200 μL of cell suspension was added to the treated dishes and allowed to adhere for 1 h at 37 °C and 5% $CO_2$. The attached cells were fixed in 4% paraformaldehyde solution for 15 min. After washing three times with PBS, the cells were permeabilized in 0.1% Triton X-100 solution for 2 min. After washing once with PBS, the cells were blocked in 3% BSA solution for 1 h. Next, the cells were incubated with anti $\alpha$-tubulin primary antibody solution (ab7291, Abcam) in 3% BSA for 1 h. After washing three times with PBS, the cells were incubated with Alexa Fluor 647 conjugated secondary antibody solution (A-21235, Thermo Fisher) in 3% BSA for 1 h. Finally, after washing three times with PBS, the samples were stored in PBS at 4 °C before use.

### 2.1.2. Imaging and Data Processing

For the reversible photoswitching of Alexa Fluor 647, an imaging buffer (10% glucose, 10 mM NaCl, 50 mM Tirs, 500 μg/mL glucose oxidase, 40 μg/mL catalase, 100 mM mercaptoethylamine, pH 8.0) was used. The samples were imaged on an Olympus IX73 inverted microscope equipped with a 60×/NA1.42 oil-immersion objective (Olympus, Tokyo, Japan). A 640 nm laser (LaserWave, Beijing, China) was controlled to have an intensity of ~5 kW/cm$^2$ at the sample plane to excite the Alexa Fluor 647. The fluorescence emission was collected by the oil-immersion objective, transmitted by a filter (FF01-680/42, Semrock, West Henrietta, NY, USA), and captured by a Hamamatsu Flash 4.0 v3 camera with an exposure time of 10 ms. Finally, we obtained the localizations table by processing the raw image with QC-STORM, which is a fitting-based method for fast multi-emitter localization [24].

### 2.2. *Simulated Datasets*
### 2.2.1. Simulated Datasets for the Performance Comparison of FACAM with ClusterViSu

We downloaded the DBSCAN codes from Github (https://github.com/imaginespark/clustering accessed on 18 September 2022) and used them to analyze the localization table of CD138. Then, we processed the clustering result into a Ground-truth dataset containing 16 clusters in a field of view of 2 μm × 1.8 μm. In other words, Ground-truth was derived from the clustering results of the experimental data by DBSCAN. Finally, we produced three simulated datasets by adding different densities of random noise to the Ground-truth. Each random noise localization was assigned uncertainties (localization precision), which were sampled randomly from a real SMLM dataset. Here, we chose F1-measure as the evaluation criterion for comparing the performance of FACAM and ClusterViSu. The F1-measure was the harmonic mean of precision (*P*) and recall (*R*), and is defined as:

$$F1 = \frac{2 \times PR}{P + R} \tag{1}$$

*P* and *R* are calculated as:

$$P = \frac{TP}{TP + FP}, \; R = \frac{TP}{TP + FN} \tag{2}$$

where *TP* (true positive) represents the number of clusters that belong to the Ground-truth, *FP* (false positive) represents the number of clusters that are not clustered in the Ground-truth, and *FN* (false negative) represents the number of clusters that do not belong to the Ground-truth.

### 2.2.2. Simulated Datasets for the Performance Comparison of FACAM with Ripley's H Function

We used ThunderSTORM [25] to generate three simulated datasets. They both had a field of view of 2 μm × 2 μm. The first one contained eight circular clusters of 100 nm radius. The second contained circular clusters with radii of 100 nm, 90 nm, 80 nm, 70 nm, 60 nm, 50 nm, 40 nm, and 30 nm, respectively. The third one contained irregular clusters. These simulated datasets were used to compare the performance of the clustering algorithms.

### *2.3. FACAM*
### 2.3.1. The Working Principle of FACAM

The working principle of FACAM is to apply the reported Alpha Shape algorithm to a selected SMLM dataset, and to determine the segmentation threshold adaptively. Alpha Shape is a popular algorithm for extracting point cloud boundaries and has been used in many different research fields [26–28]. In fact, Nerreter et al. tried to use Alpha Shape in SMLM [29], although they did not provide sufficient details in that paper.

We noticed that Alpha Shape can be quickly implemented by Delaunay triangulation [30]. For a given planar point set S and parameter $\alpha$, the implementation can be performed as follows. First, a Delaunay triangulation is constructed for the point set S. If the length of any edge in a triangle is greater than $2 \times \alpha$, the triangle is deleted. Then, for each edge in the remaining triangles, if at least one of the two circles (which have a radius of $\alpha$ and pass through the two endpoints of each edge) does not have a point that belongs to the point set S, the triangle is kept. Otherwise, the triangle is deleted. Finally, the edges of the retained triangle network are found, and the endpoints of the edge lines are the boundary points.

Figure 1 illustrates a dynamic clustering process as the segmentation threshold ($\alpha$) increases. At a low segmentation threshold ($\alpha1$ or $\alpha2$), the clustering results were far from the Ground-truth in Figure 1a. At a suitable segmentation threshold ($\alpha3$), the clustering is accurately matched with the Ground-truth. However, at a high segmentation threshold ($\alpha4$), the clustering results were contaminated with noise. Therefore, to guarantee an accurate clustering, we needed to determine an appropriate segmentation threshold.

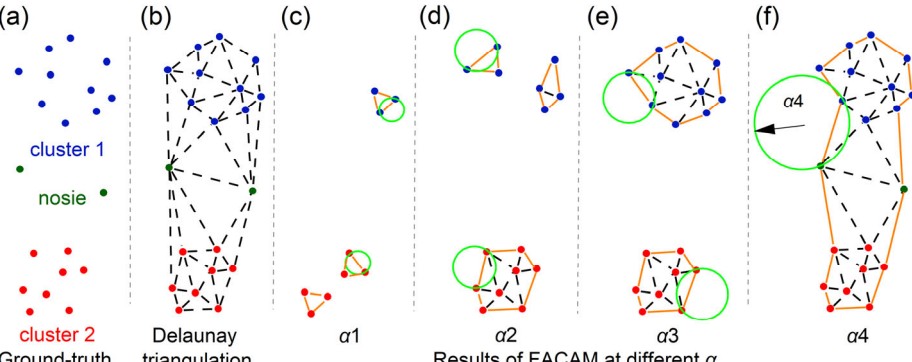

**Figure 1.** The working principle of FACAM. (**a**) A point set containing two clusters. (**b**) Construction of Delaunay triangulation for the point set in (**a**). (**c–f**) Clustering results of FACAM at different segmentation thresholds.

### 2.3.2. Adaptive Determination of Segmentation Threshold

The localizations in the SMLM frequently exhibit high aggregation because of the blinking characteristics of fluorescent dyes and a repetitive localization of a single blinking event [31]. To obtain an accurate setting on the value of $\alpha$ automatically, here we took advantage of the average localization precision ($\sigma$) in a localization table, and picked out the positions of core localizations from the localization table. A localization can be defined as a core localization, only if we can find at least two additional localizations within a radius of $2\sigma$. The reason is simple: the farthest distance between two localizations (see the blue localizations in Figure 2b) should be $2\sigma$. Then, we linked the localization precision of core localizations to the mean nearest neighbor distance (*MNND*). We took the average of the localization precision and the *MNND* to reduce the effect of variability in the local density. Therefore, the segmentation threshold was determined as:

$$\alpha = \sqrt{MNND^2 + \sigma^2} \tag{3}$$

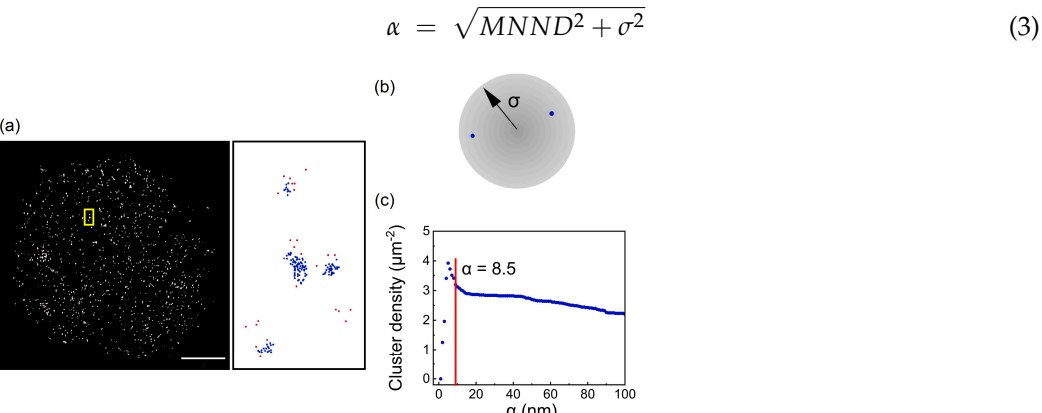

**Figure 2.** Adaptive determination of the segmentation threshold in FACAM. (**a**) A reconstructed super-resolution image of CD138. The dotted image is from the enlarged image of the rectangular region (yellow). Moreover, blue dots represent core localizations in the dotted image. Scale bar: 4 μm. (**b**) Single molecule localization with localization precision ($\sigma$). (**c**) The dependence of clustering density and segmentation threshold. Here the segmentation threshold ($\alpha$ = 8.5 nm) was calculated by FACAM.

Taking an experimental dataset of CD138 as an example, we illustrated the principle of adaptive determination of the segmentation threshold in FACAM (Figure 2). Importantly, we found that the segmentation threshold should not be set at the maximum clustering density. This is also explained in Figure 1. At a segmentation threshold ($\alpha$1 or $\alpha$2), we obtained the maximum clustering density. However, neither $\alpha$1 nor $\alpha$2 was a suitable segmentation threshold.

### 2.4. Counting the Number of Protein Molecules in a Cluster

Counting the number of protein molecules has been an important goal in quantitative SMLM. Hummer et al. reported a formula to describe the distribution of individual molecule blinking events (Equation (4)) [32]. Here we could use FACAM to extract the distribution of blinking events within a single cluster, and then perform a fit to Equation (4). Based on the fact that each nuclear pore complex includes 32 copy of proteins [6], we tested the accuracy of the formula by treating a complete nuclear pore complex as a 32-mers.

$$Pm(n) = \sum_{k=0}^{\min(m,n)} \binom{m}{k} \binom{n}{k} q^{m-k} (1-q)^k p^{k+1} (1-p)^{n-k} \tag{4}$$

where $p$ is the proportion of molecules that do not blink after initial light activation, $q$ represents the proportion of undetected molecules, and n refers to the number of blinking events (the number of burst events minus one) within a single cluster. For $m$ = 0, 1, 2, *Pm*

(*n*) represents the distribution of the number of blinking events for monomers, dimers, and trimers, respectively.

## 3. Results and Discussion

### 3.1. The Clustering Performance of FACAM, DBSCAN, and ClusterViSu on Simulated Data with Different Noise Levels

We compared the performance of FACAM with DBSCAN and ClusterViSu on the simulated data, and used F1-measure as a metric. It should be noted that a higher F1-measure value indicates that the clustering results are closer to Ground-truth, and an F1-measure value of 1 means a perfect match. We obtained different simulated datasets by adding different levels (or densities) of random noise (low noise: 200 localizations/$\mu m^2$; moderate noise: 400 localizations/$\mu m^2$; high noise: 1000 localizations/$\mu m^2$) to the Ground-truth dataset. Each localization contained a positional uncertainty sampled from a real SMLM experiment. The reconstructed super-resolution images of the Ground-truth and the simulated datasets are shown in Figure 3a,b, respectively.

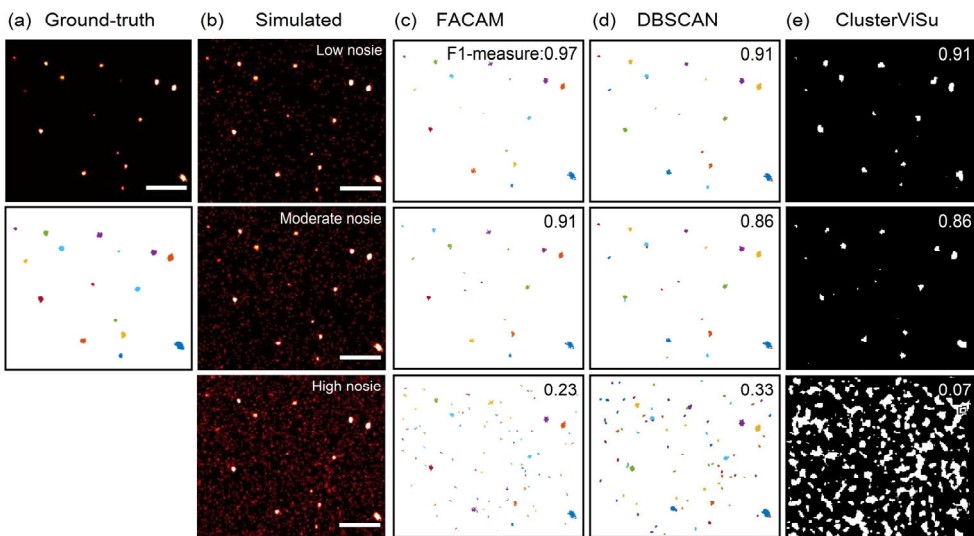

**Figure 3.** The clustering performance of FACAM, DBSCAN, and ClusterViSu on the simulated dataset. (**a**) Ground-truth (top) and its dotted image (bottom). (**b**) Reconstructed super-resolution images of the simulated datasets containing different levels of noise. The noise levels in (**b**): (top) low noise, (middle) moderate noise, (bottom) high noise. (**c**) The clustering results from FACAM for the simulated datasets in (**b**). (**d**) The clustering results from DBSCAN for the simulation datasets in (**b**). (**e**) The clustering results from ClusterViSu for the simulation datasets in (**b**). F1-measure values are also shown at the top-right corners of each figure in (**c**–**e**). Scale bar: 500 nm.

As shown in Figure 3c–e, both FACAM, DBSCAN (MinPts = 2, $\varepsilon$ = 10 nm), and ClusterViSu exhibited good noise immunity on the simulated datasets with low and moderate noise, but FACAM had a slightly better clustering performance than DBSCAN and ClusterViSu. In the simulated dataset with high noise, the clustering performance of FACAM, DBSCAN, and ClusterViSu was poor, although DBSCAN provided a better F1-measure value than FACAM and ClusterViSu. As we can also see in this high noise case, ClusterViSu identified many large clusters, indicating that ClusterViSu was unable to obtain accurate segmentation thresholds in this kind of high noise scenario. Instead, FACAM and DBSCAN were able to obtain clusters with sizes closer to the Ground-truth. Based on these findings, we believe FACAM provides a more accurate clustering than ClusterViSu. By adaptively determining the segmentation threshold, FACAM can achieve a similar performance to DBSCAN. Moreover, as shown in Figure 3c–e, FACAM and DBSCAN were able to distinguish different clusters using different colors, while ClusterViSu only

showed pixel-based binarized images. That is to say, the clustering results from FACAM and DBSCAN were easier to visualize.

### 3.2. The Clustering Performance of FACAM, DBSCAN, and ClusterViSu on Simulated Data

We compared the performance of FACAM with DBSCAN (MinPts = 3, $\varepsilon$ = 10 nm) and ClusterViSu on three different simulated datasets. As shown in Figure 4, the first simulated dataset contained eight circular clusters with a radius of 100 nm. In the results from FACAM, DBSCAN, and ClusterViSu, the mean value of the radius was 98.6 nm, 97.4 nm, and 84.2 nm, respectively. FACAM obtained more accurate results than DBSCAN and ClusterViSu. For the second simulated dataset, which contained eight circular clusters with a radius changing from 100 nm to 30 nm, the results of FACAM and DBSCAN were basically the same, and the result of ClusterViSu was still smaller. Finally, in the third simulated dataset that contained irregular clusters, FACAM and DBSCAN exhibited a better performance than ClusterViSu. This was because the shape change of ClusterViSu was greater than FACAM and DBSCAN. In summary, for the quantification of heterogeneous and irregular clusters, FACAM and DBSCAN exhibit better performance than ClusterViSu. FACAM performs similarly or even better than DBSCAN without the need for user-defined parameters.

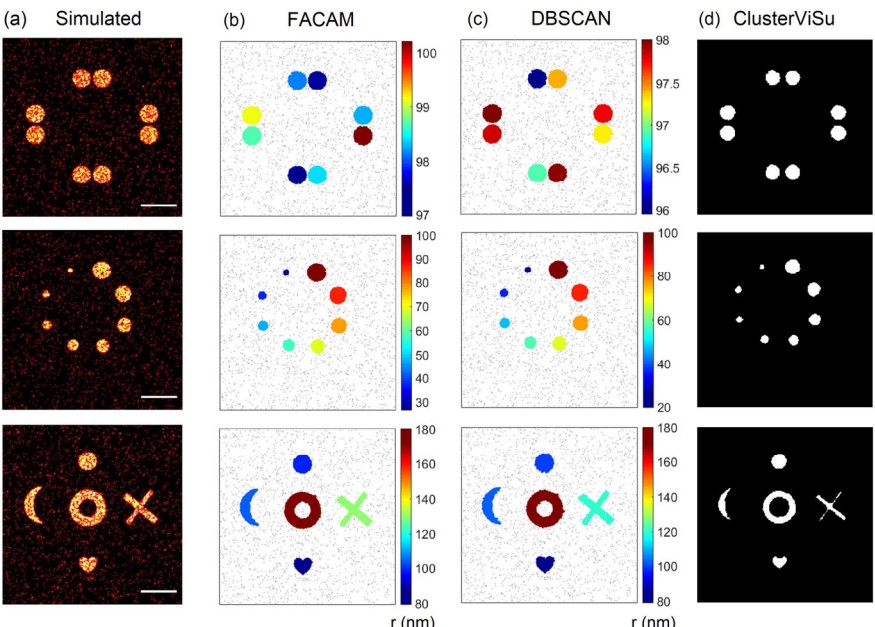

**Figure 4.** The clustering performance of FACAM, DBSCAN, and ClusterViSu on simulated datasets. (**a**) Reconstructed super-resolution images of the simulated datasets. Scale bar: 400 nm. (**b**) The clustering results from FACAM for the simulated datasets in (**a**). (**c**) The clustering results from DBSCAN for the simulated datasets in (**a**). (**d**) The clustering results from ClusterViSu for the simulated datasets in (**a**). Color bar = radius (nm).

### 3.3. The Computation Time of FACAM and Several Popular Clustering Analysis Methods

We calculated the computation time of FACAM and several popular clustering analysis methods (DBSCAN (MinPts = 5, $\varepsilon$ = 10 nm), Ripley's H function, ClusterViSu) using SMLM datasets of different sizes. All programs were implemented on a desktop computer with an Intel i7-9700 CPU at 3.00 GHz and 16 GB of RAM. Furthermore, all programs were running on the same CD138 data, which contained more than one million localizations (Figure 5a). We controlled the number of localizations by choosing the ROI. As shown in Figure 5b, FACAM was able to process millions of localizations in half an hour, which was more than ten times faster than DBSCAN (7 h). The Ripley's H function, which cannot provide quantitative information (including the number, location, and shape of the protein clusters), took a relatively small amount of time (about 3 h). However, we could not obtain the computation time of ClusterViSu, because this method was not capable of processing

such a big dataset. For a smaller dataset (tens of thousands of localizations), ClusterViSu took more than a few minutes to process, which was much slower than the other three clustering methods (Figure 5). Note that ClusterViSu uses Monte Carlo simulations in the determination of the segmentation threshold, which occupies most of the computation time. Moreover, as shown in Table 1, when the number of localizations was $1 \times 10^5$, FACAM had a six times speed advantage over DBSCAN. In addition, in the range of greater than $2 \times 10^5$ localizations, FACAM had more than a ten times speed advantage over DBSCAN. In general, FACAM is faster than DBSCAN and ClusterViSu.

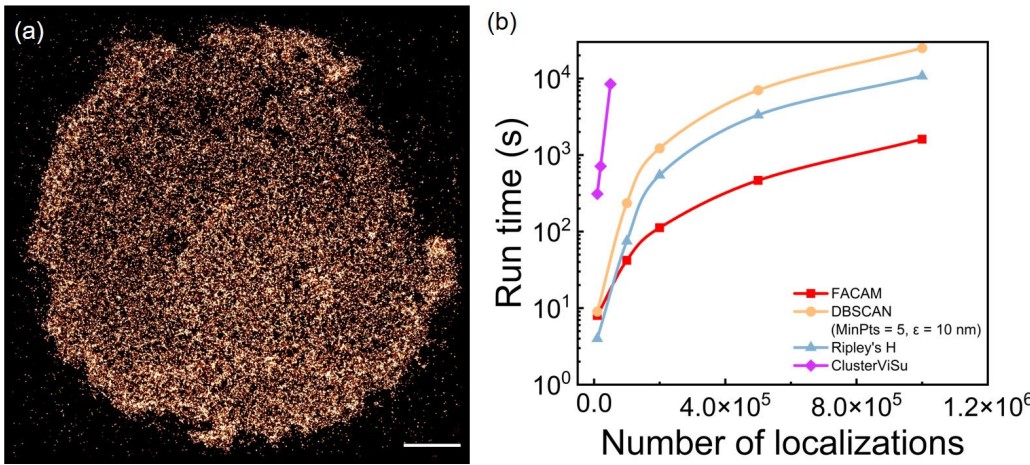

**Figure 5.** The dependence of computation time on the number of localizations. (**a**) The reconstructed super-resolution images of CD138. (**b**) Comparison of computation speed between FACAM, DBSCAN, ClusterViSu, and Ripley's H function. Note that the vertical coordinates are logarithmic (base 10).

**Table 1.** The computation time of different clustering algorithms at different numbers of localizations.

| Number of Localizations | FACAM | DBSCAN | Ripley | ClusterViSu |
|---|---|---|---|---|
| $1 \times 10^4$ | 8 s | 9 s | 4 s | 325 s |
| $1 \times 10^5$ | 42 s | 235 s | 75 s | * |
| $2 \times 10^5$ | 112 s | 1220 s | 545 s | * |
| $5 \times 10^5$ | 465 s | 7020 s | 3327 s | * |
| $1 \times 10^6$ | 1609 s | 24,955 s | 10,750 s | * |

\* Means that the computation time is not available.

### 3.4. Quantitative Analysis of Experimental Data by FACAM, DBSCAN, and ClusterViSu

We performed a quantitative analysis on the experimental SMLM data of CD138, after clustering with FACAM, DBSCAN (MinPts = 3, $\varepsilon$ = 20 nm), and ClusterViSu. The results of FACAM are shown in Figure 6d. We calculated the area of the Delaunay triangulation for each cluster, and therefore obtained the equivalent radius of this area. The distribution of the equivalent radius of each cluster is normally used to quantify the cluster size. We found that the equivalent radius of most of the clusters was less than 40 nm. Then, we further carried out a nearest neighbor analysis (NeNA) on the clusters to obtain the spatial distribution, and found that the distance between clusters was mostly in the range of 100–500 nm. These indicate that the clusters were small and well-separated from each other. In addition, we counted the distribution of the number of localizations within a single cluster, which could be used to extract internal information about the clusters. Finally, we counted the distribution of the blinking events within the cluster that are used to calculate the number of proteins. However, neither ClusterViSu nor DBSCAN provided information about the distribution of the blinking events. For this CD138 data, FACAM and DBSCAN obtained almost the same number of clusters (423, 418, respectively) while ClusterViSu obtained 703. This led to the difference in the distribution of quantitative information

between ClusterViSu and FACAM or DBSCAN. As shown by the red arrows in Figure 6c,d, it should be noted that the minimum number of localizations within a single cluster of DBSCAN depends on the parameter MinPts. This is one of the main factors contributing to the difference in the distribution of the results between DBSCAN and FACAM.

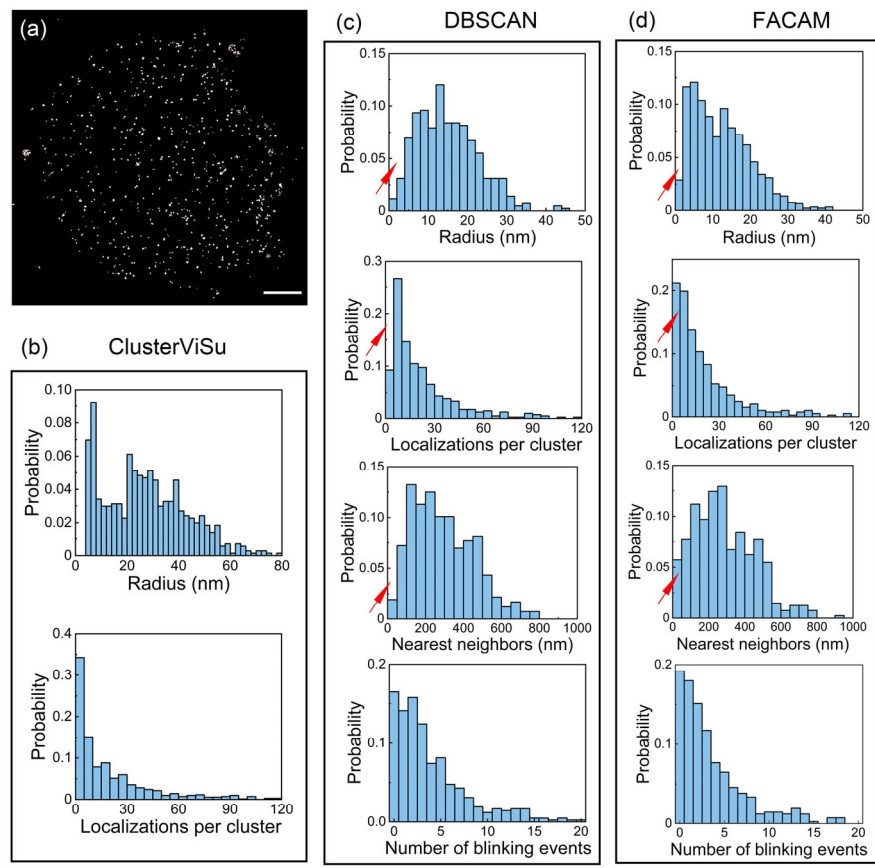

**Figure 6.** Quantitative analysis on the clustering results from FACAM. (**a**) A reconstructed super-resolution image of the distribution of CD138 on a RPMI-8226 cell. Scale bar: 2μm. (**b**) The results of ClusterViSu in (**a**). Here we obtained 703 clusters. (**c**) The results of DBSCAN in (**a**). Here we obtained 418 clusters. (**d**) The results of FACAM in (**a**). Here we obtained 423 clusters. The red arrows point out main differences in the distribution of results between DBSCAN and FACAM.

### 3.5. Validation of the Clustering Performance Using Prior Biological Knowledge
3.5.1. Validation via Nuclear Pore Complexes

Nuclear pore complexes (NPCs) are commonly used to validate quantitative microscopy because of their specific spatial structures. Here, we used the SMLM data of Nup96-GEP nanobody-X4-AF647 (downloaded from Thevathasan's paper [6]). A schematic top view of the Nup96 structure is shown in the lower-left of Figure 7a. It was clear that a complete Nup96 structure contained eight corners, and SMLM was able to visualize some of the corners (Figure 7b). From previous SMLM studies [6], it is well accepted that the closest distance between the corners is 42 nm, while the farthest distance is 107 nm. We performed the clustering analysis on the Nup96 data using FACAM, ClusterViSu, and DBSCAN. The clustering results of FACAM are shown in Figure 7b–e. We obtained the distribution of nearest neighbors between the corners (Figure 7d). Using a Gaussian fit, we found that the nearest distance between the corners was 42.0 nm. This is exactly the same as the reported value. It should be noted that, due to the characteristics of the data, using an adaptive segmentation threshold can only obtain clusters of corners. Therefore, we increased the segmentation threshold to obtain clusters of the complete Nup96 structures (Figure 7c). We further performed a circle fit to single Nup96 structures to provide the

diameter distribution shown in Figure 7e. The diameter of Nup96 was calculated to be 111.3 nm, which is 4.3 nm larger than the reported value of 107 nm by Thevathasan et al. [6].

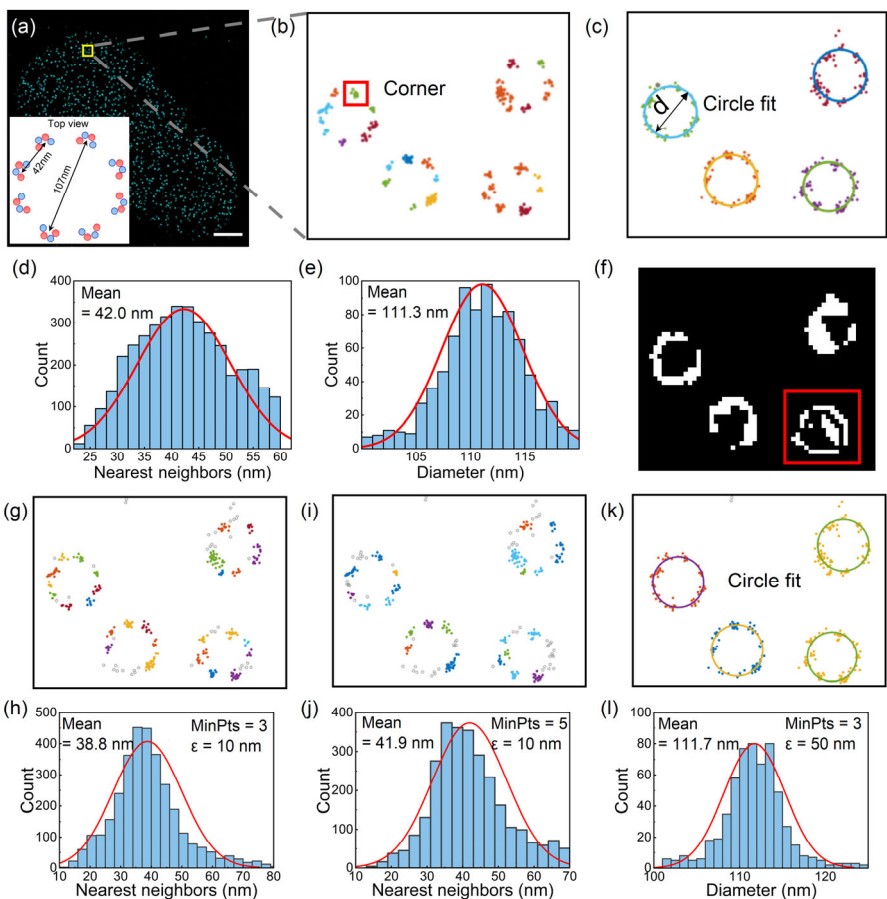

**Figure 7.** Validating the clustering performance of FACAM, Clusters, and DBSCAN on nuclear pore complexes. (**a**) A super-resolution image and a schematic top view (bottom left) of Nup96 structures. Scale bar: 1 μm. (**b–e**) The clustering result of FACAM. (**b**) The clustering results of Nup96 in the enlarged region in (**a**), using FACAM and adaptive segmentation threshold. (**c**) The clustering results of Nup96 in the enlarged region in (**a**), using FACAM and artificial segmentation threshold. (**d**) The distribution of nearest neighbors between corners. (**e**) The diameter distribution of Nup96. (**f**) The clustering results of Nup96 in the enlarged region in (**a**), using ClusterViSu. The red box shows the artifacts appeared inside the Nup96 structures. (**g–l**) The clustering result of DBSCAN. (**g,h**) The clustering results of Nup96 in the enlarged region in (**a**) and the distribution of nearest neighbors between corners, using DBSCAN (MinPts = 3, ε = 10 nm). (**i,j**) The clustering results of Nup96 in the enlarged region in (**a**) and the distribution of nearest neighbors between corners, using DBSCAN (MinPts = 5, ε = 10 nm). (**k,l**) The clustering results of Nup96 in the enlarged region in (**a**) and the diameter distribution of Nup96, using DBSCAN (MinPts = 3, ε = 50 nm).

The clustering results of ClusterViSu for the same dataset are shown in Figure 7f. We found that ClusterViSu was unable to segment out the corners, and that some artifacts appeared inside the Nup96 structures, as shown in the red box in Figure 7f. Moreover, the pixelated binary image from ClusterViSu was not able to show the structural features of Nup96. These findings indicate that ClusterViSu may not be suitable for segmenting hollow clusters such as NPCs. As for DBSCAN, the clustering results are shown in Figure 7g–l. With the parameters MinPts = 3, ε = 10 nm, and MinPts = 5, ε = 10 nm, we obtained that the nearest distances between the corners were 38.8 nm and 41.9 nm, respectively (Figure 7h,j). This shows that the choice of parameters affects the accuracy of DBSCAN. Finally, with the parameters MinPts = 3 and ε = 50 nm, we obtained that the diameter of Nup96 was

111.7 nm, which is 4.7 nm larger than the reported value of 107 nm. The above indicates that with the right parameters, DBSCAN is able to obtain essentially the same results as FACAM.

### 3.5.2. Validation via Membrane Protein CD138

We performed SMLM experiments on a common membrane protein (CD138) of RPMI-8226 cells. Nerreter et al. showed the variation in the clustering density of CD19 at different antibody concentrations [29]. We used a similar antibody concentration setting to obtain the dependence of the cluster density on the antibody concentration. The six different antibody concentrations were $1.07 \times 10^{-4}$ μg/mL, $1.07 \times 10^{-3}$ μg/mL, $1.07 \times 10^{-2}$ μg/mL, $1.07 \times 10^{-1}$ μg/mL, 2.675 μg/mL, and 5.35 μg/mL, respectively. Immunofluorescence labeling was used to label the membrane protein with Alexa Fluor 647. We collected ten sets of SMLM data at each concentration, and presented representative super-resolution images for each concentration in Figure 8a–f. We further performed a clustering analysis on these datasets using FACAM, DBSCAN (MinPts = 3, $\varepsilon$ = 20 nm), and ClusterViSu. Figure 8g shows the dependence of the cluster density on the antibody concentration. We found that the cluster density increased linearly with the antibody concentration in a wide range, and was saturated at an antibody concentration of about 1μg/mL or higher (speculated from the fitting curve). Interestingly, FACAM presented a better linearity ($R^2$ = 0.99) in the fitting than ClusterViSu ($R^2$ = 0.95) and DBSCAN ($R^2$ = 0.97).

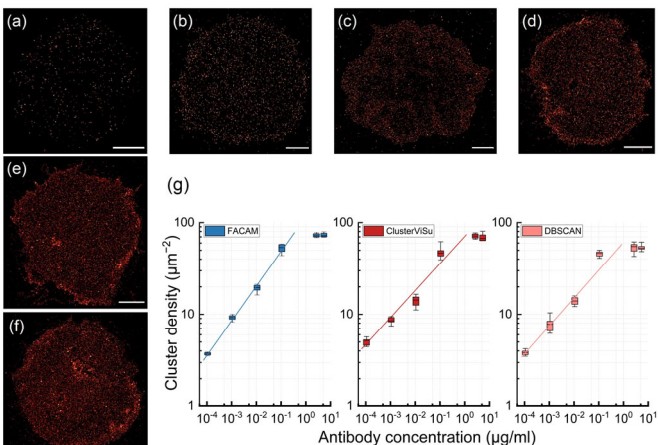

**Figure 8.** The dependence of cluster density on antibody concentration. (**a**–**f**) Reconstructed super-resolution images of CD138 at different antibody concentrations. From (**a**) to (**f**): $1.07 \times 10^{-4}$ μg/mL, $1.07 \times 10^{-3}$ μg/mL, $1.07 \times 10^{-2}$ μg/mL, $1.07 \times 10^{-1}$ μg/mL, 2.675 μg/mL, and 5.35 μg/mL, respectively. Scale bar: 4 μm. (**g**) The dependence of cluster density on antibody concentration, using FACAM (blue) and ClusterViSu (red) as the clustering method. Boxplots were used for the statistics, and a linear fit to the median values of the first four boxes is performed. Note that the coordinates are both logarithmic (base 10).

### 3.5.3. Validation via Erythrocyte Cytoskeletons

SMLM has been used to determine the junction-to-junction distance in erythrocyte cytoskeletons. Pan et al. [22] determined this distance in erythrocyte cytoskeletons as ~80 nm. We used this biological prior knowledge to verify the accuracy of FACAM. With the help of FACAM and DBSCAN, we performed a quantitative analysis on the distribution of N termini β-spectrin on an erythrocyte cytoskeleton (see Figure 9a for a representative super-resolution image). Figure 9b,c show the clustering results of FACAM. We visualized the clustering results of the intact erythrocyte skeleton for the N termini of β-spectrin (Figure 9b). Then, we analyzed the distance distribution of nearest neighbors between the clusters. Using a Gaussian fit, we obtained a mean distance of 76.9 nm, which is close to the value (~80 nm) by Pan et al. [22]. For the clustering results of DBSCAN (Figure 9d–f), with

the parameters MinPts = 3, $\varepsilon$ = 20 nm, and MinPts = 5, $\varepsilon$ = 20 nm, we obtained the distances of 81.5 nm and 85.8 nm, respectively. Although the DBSCAN results were all close to 80 nm, the user-defined parameters directly affected the final result. However, FACAM was able to obtain similarly accurate results by adaptively determining the segmentation threshold.

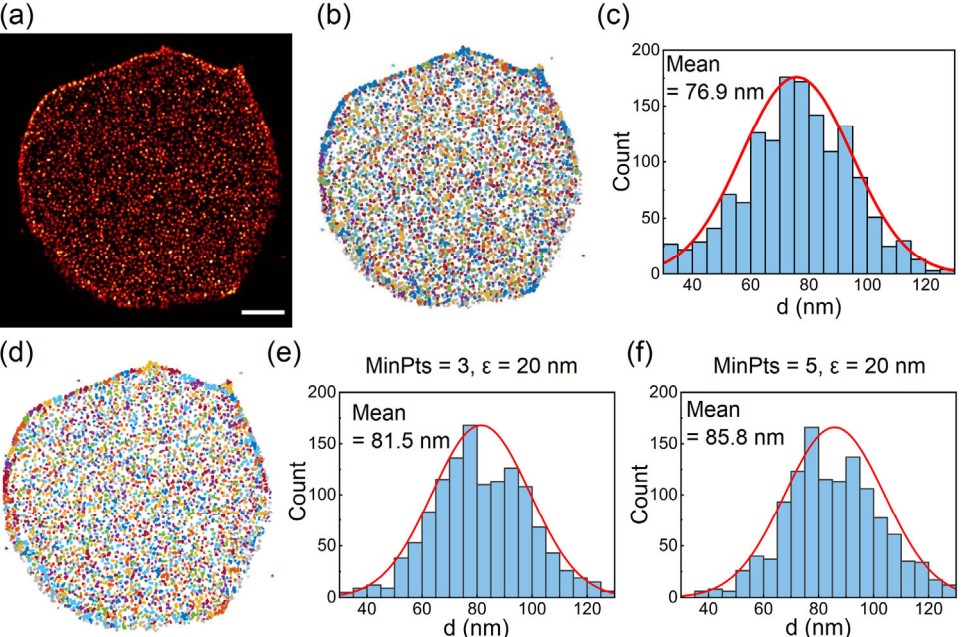

**Figure 9.** Comparison of FACAM and DBSCAN in determining the junction-to-junction distance in the erythrocyte cytoskeleton. (**a**) A reconstructed super-resolution image showing the distribution of β-spectrin in the erythrocyte cytoskeleton. Scale bar: 1 μm. (**b**) Visualization of the clustering results from FACAM in (**a**). A total of 1459 clusters were obtained. (**c**) The distribution of nearest neighbor distance between clusters from FACAM. $R^2$ = 0.99. (**d**) Visualization of the clustering results from DBSCAN in (**a**). A total of 1468 clusters were obtained (MinPts = 3, $\varepsilon$ = 20 nm). (**e**) The distribution of nearest neighbor distances between clusters from DBSCAN (MinPts = 3, $\varepsilon$ = 20 nm). $R^2$ = 0.96. (**f**) The distribution of nearest neighbor distances between clusters from DBSCAN (MinPts = 5, $\varepsilon$ = 20 nm). $R^2$ = 0.95.

### 3.6. Protein Counting

The purpose of the quantitative analysis of SMLM data is to extract information such as the density, size, distribution, shape, and the number of proteins [4]. Previously, we showed the application of FACAM in obtaining the quantitative information about protein complexes such as the size, density, and the distribution of nearest neighbors. Therefore, in this chapter, we showed the application of FACAM in protein counting. We evaluated the performance of FACAM in the stoichiometry of monomers and dimers. The principle of protein counting is described in Section 2.4. We obtained SMLM data from Baldering et al. [23], and verified the oligomerization status of the data using the monomer (CD86-mEos4b) and the dimer (CTLA4-mEos4b). Figure 10a–c shows the process of obtaining the number of blinking events, and Figure 10d–g present the analysis results (CD86-mEos4b: $p$ = 0.27; CTLA4-mEos4b: $p$ = 0.27, $q$ = 0.35), which are consistent with the literature [23].

In addition, we regarded a single nuclear pore complex as a 32-mers and performed an interesting experiment to further validate Equation (4). We carried out clustering in the NPC data (Figure 10h) using FACAM, and fitted the corresponding distribution in Figure 10i to obtain the protein counting information ($p$ = 0.23, $q$ = 0.29). Here, the difference between 1 and the $q$ value (1-$q$ = 0.71) is actually consistent with the effective labeling efficiency of 73 $\pm$ 7% reported in Thevathasan's paper [6].

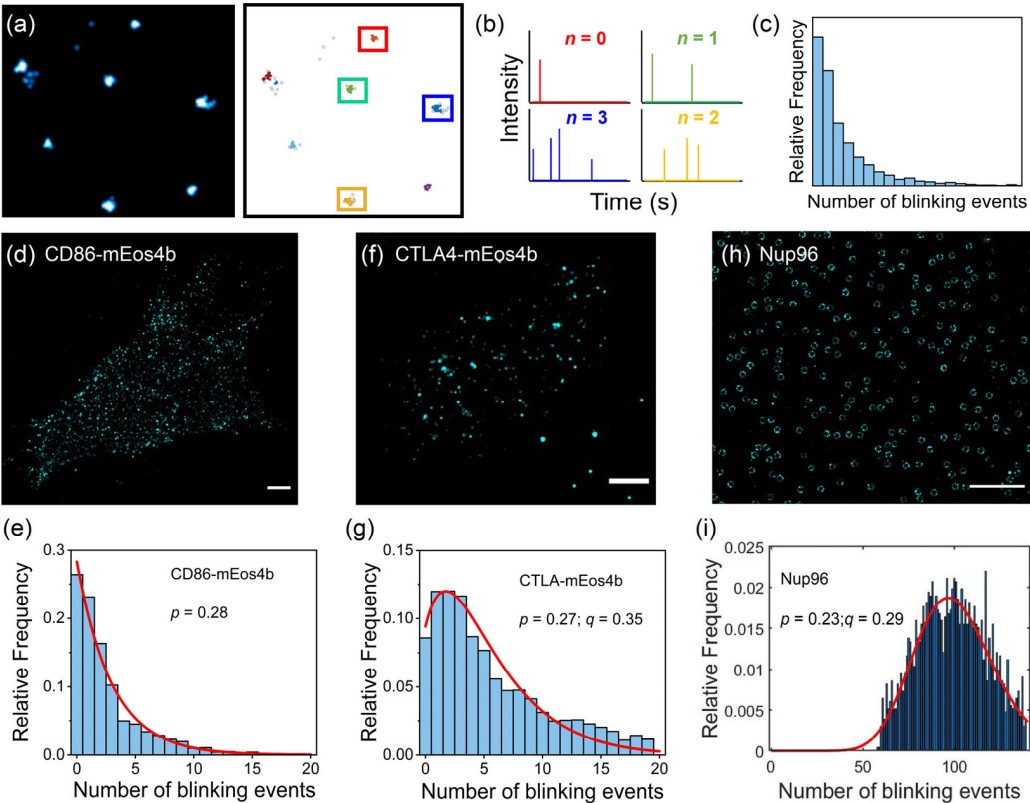

**Figure 10.** Applying FACAM in counting protein molecules. (**a–c**) The process of determining the number of blinking events in an intra-cluster. (**a**) A reconstructed super-resolution image of mEos4b and its clustering result using FACAM. (**b**) Schematic intensity time traces showing the number of blinking events ($n$) of single mEos4b molecules. The different color box contains the different clusters. (**c**) Relative frequency of the number of blinking events of mEos4b molecules. (**d,e**) A reconstructed super-resolution image of CD86-mEos4b and the corresponding distribution of the number of blinking events in the intra-cluster. $R^2$: 0.98. The number of clusters is 1250. (**f,g**) A reconstructed super-resolution images of CTLA4-mEos4b and the corresponding distribution of the number of blinking events in the intra-cluster. $R^2$: 0.95. The number of clusters is 3275. Clusters with irregular shapes and overlapping clusters were filtered out. (**h,i**) A reconstructed super-resolution image of Nup96 viewed as 32-mers and the corresponding distribution of the number of blinking events in the intra-cluster. $R^2$: 0.93. The number of clusters is 2319. Clusters that are heavily vestigial and overlapping are filtered out. Scale bar: 2 μm (**b,d**), 1 μm (**f**).

## 4. Conclusions

This paper presents a new clustering method called FACAM, and aims to solve a main challenge in current clustering methods, that is, a slow running speed. Using simulated and experimental data, we proved that FACAM is able to extract quantitative information about protein complexes (including the size, density, distribution of nearest neighbors, and the number) accurately. More importantly, using parallel computing, FACAM is able to process millions of localizations in less than an hour, which is ten times faster than DBSCAN, thus enabling a daily use of clustering. We further verified the applications of FACAM in SMLM data with different structures (membrane proteins, erythrocyte backbone proteins, and nuclear pore complexes). By comparing the simulated and experimental data, FACAM outperformed ClusterViSu [17]. In addition, with adaptively determined segmentation thresholds, FACAM obtained similar or even better results than DBSCAN [12]. In fact, FACAM can process more biological structures whose SMLM data can form clusters, such as mitochondria. Unfortunately, we have not extended FACAM to handle 3D SMLM datasets. We believe this work is helpful in pushing forward the use of quantitative SMLM for studying important biological questions.

**Author Contributions:** Conceptualization, C.W.; methodology, Z.Z.; data curation, W.K.; software, analysis validation and writing—original draft preparation, C.W.; writing—review and editing, C.W., Y.Z. and Z.-L.H.; supervision, Z.-L.H. All authors have read and agreed to the published version of the manuscript.

**Funding:** National Natural Science Foundation of China (82260368, 81827901) and Start-up Fund from Hainan University (KYQD(ZR)-20077).

**Informed Consent Statement:** Not applicable.

**Data Availability Statement:** Data underlying the results presented in this paper are not publicly available at this time but may be obtained from the authors upon reasonable request.

**Acknowledgments:** We thank the Optical Bioimaging Core Facility of WNLO-HUST for technical support.

**Conflicts of Interest:** The authors declare no conflict of interest.

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
