# Peer review of "FACAM: A Fast and Accurate Clustering Analysis Method for Protein Complex Quantification in Single Molecule Localization Microscopy"

_photonics, doi:10.3390/photonics10040427_

Round 1
Reviewer 1 Report
In this work, the authors significantly improve the computation time for clustering analysis by modifying the Alpha Shape method. This modification takes into account not only molecule localizations but also localization precision parameter, which has not been considered previously. The inclusion of the latter allows for the adapted threshold and thereby alleviates possible bias caused by subjective threshold setting. The improvement of the computation time is documented by comparison with other available and widely used clustering methods such as DBSCAN, ClusterViSu and Ripley’s H function. Importantly, the real world use of the method is demonstrated on the previously available data on the several biological samples, which provide insight into the organization of membrane proteins, cytoskeleton and nuclear pores.
General comments:
Introduction is nicely structured, brief but extensive in listing the appropriate clustering approaches available so far together with their pros and cons. I’m giving thumbs up that already published data were used together with the experimental data produced in this work. Material and methods are sufficient and Results are clearly presented. I have some suggestions below for adding a bit of more data. Last section, Conclusions, should include however more discussion summarizing why and how FACAM outperforms previous clustering methods.
Comments:
Line 31: Please, could you add to the sentence "...important questions in cell biology [1], such as gene transcription (Hoboth et al., IJMS 2021) or interorganelle contacts (Nieto-Garai et al., IJMS 2022)."
Line 41: Please, briefly add which "certain characteristics", incl. appropriate reference(s).
Lines 72-73: “ClusterViSu can only analyze ROIs with a maxi-72 mum size.” What maximum size?
Why the authors choose to present in the Results section 3.1 the comparison of FACAM with ClusterViSu and not (also) with DBSCAN? Wouldn’t be appropriate to compare FACAM in addition to ClusterViSu also with DBSCAN? Could the authors show the clustering performance and handling of the noise by DBSCAN in addition to FACAM and ClusterViSu and this to the Results section 3.1?
Fig. 4 / Results 3.2: I missed how was estimated that “Ripley's H function was unable to quantify these heterogeneous clusters” (Line 255-256)? Again, on what quantitative measure is based the statement that “FACAM exhibits a better performance than Ripley's H function” (Line 258)? Is it estimated from the shape of H(r)?
Fig. 7 / Results 3.5.1: How about adding DBSCAN data, similarly as shown ClusterViSu in Fig. 7f? The reason why I’m asking this and previous question is that to mee DBSCAN seems more comparable with FACAM than ClusterViSu or H(r). DBSCAN representation is in fact not pixelated as ClusterViSu is. Could you comment and/or add this data?
Line 344: Please, rephrase “We did a similar antibody concentration setting against the experiment”.
Fig. 8g: Wouldn’t be better other fitting method than linear fit, as the last data points (for the highest antibody conc.) reach plateau and are not included in the liner fit?
Fig. 9 / Results 3.5.3: Could you add the comparison with DBSCAN to show that FACAM outperforms DBSCAN also on the example of erythrocyte cytoskeleton?
Line 381: Could you use “is described” instead of “can be seen”?
Section 4. Conclusion: In my view, this section should be slightly extended, including references, to discuss the comparison with previously published clustering methods and stating the reasons why FACAM outperforms them.
Conclusion:
The improvement in the clustering analysis presented here should be of interest for the bioimage analysis community and therefore I recommend its publication in the MDPI journal Photonics with the above listed modifications of the manuscript.
Reviewer 2 Report
see attached file

Reviewer 3 Report
This manuscript is written well. This method may be used widely and generally. But, some revision should be required.
Line 122, What is QC-STORM. This definition may be important because it is regarding the processing of super-resolution images. Furthermore, it would be better to be mentioned what is the ‘localizations table’ for general readers. Furthermore, I recommend that all abbreviations should be checked.
In the legend of some figures, colors are not defined. They should be clearly mentioned.
Round 2
Reviewer 2 Report
The authors considerably improved their manuscript and added many convincing data and images that demonstrate the quality of their work. I strongly recommend to publish the manuscript and the supplement as they are.